# Undocked Tails, Mycoplasma-like Lesions and Gastric Ulcers in Slaughtering Pigs: What Connection?

**DOI:** 10.3390/ani13020305

**Published:** 2023-01-15

**Authors:** Annalisa Scollo, Mustansar Abbas, Barbara Contiero, Flaviana Gottardo

**Affiliations:** 1Department of Veterinary Sciences, University of Torino, 10095 Grugliasco, Italy; 2Department of Animal Medicine, Production and Health, University of Padova, Viale dell’Università 16, Agripolis, 35020 Legnaro, Italy

**Keywords:** tail docking, pig welfare, slaughter evaluation, mycoplasma-like lesions, gastric ulcers

## Abstract

**Simple Summary:**

Many factors contribute to the welfare and health of animals in commercial production systems and, if not well managed, might contribute to the onset of abnormal behaviors such as tail biting on pig farms. This is an expensive and welfare-decreasing complication in the current modern swine industry that might be particularly challenging in farms rearing undocked pigs. Legal and market-driven requirements of pork production from undocked pigs are increasing the percentage of animals with long tails and consequences should be evaluated. The aim of the present work was to monitor tail, pluck (lungs, pleurae, and liver), stomach, carcass, and thigh lesions in slaughtering pigs belonging either to conventional docked batches or to batches from farms with the complete ban on tail docking. Results showed a higher prevalence of tail lesions on undocked batches, suggesting that more and alternative efforts to manage long-tailed animals are needed. Moreover, undocked batches showed higher frequencies for mycoplasma-like lesions in lungs and gastric ulcers, even if it is still not clear whether tail lesions share the same predisposing factors of lung lesions and gastric ulcers, or whether tail lesions might have a role in the causality and onset of the other conditions.

**Abstract:**

Tail biting is an economical and behavioral problem in the pork production system worldwide and systematic tail docking has been applied for decades to decrease the risk of its onset. However, legal and market-driven requirements are leading pig producers to rear undocked animals. The aim of this work was to monitor tail, pluck (lungs, pleurae, and liver), stomach, carcass, and thigh lesions in slaughtering pigs belonging to either docked or undocked batches. A total of 525 batches were evaluated at slaughter: 442 docked and 83 undocked batches. The presence of tail lesions was only recorded in undocked batches (44.0 ± 0.402 vs. 0.2 ± 0.2% compared to docked ones, *p* < 0.001), with a prevalence of severe chronic lesions of 27.3% ± 0.032, suggesting that more and alternative wide efforts to manage long-tailed animals are needed. On the contrary, docked animals showed more frequent ear lesions (9.6% ± 0.037 vs. 4.6% ± 0.019; *p* = 0.0001). Severe lung lesions were found more frequently in undocked animals (9.2% ± 0.043 vs. 6.6% ± 0.011, *p* = 0.006), as well as gastric ulcers (26.1% ± 0.021 vs. 20.3% ± 0.37, *p* = 0.006). These lesions might share the same predisposing factors of tail lesions; the latter might be investigated as an iceberg indicator for other pathological conditions in undocked pigs and eventual causal association among lesions in these organs should be explored.

## 1. Introduction

There are 70 billion animals raised annually worldwide for milk, meat, and eggs. On average, 66% animals are reared intensively [1]. With growing debate over how animal-derived food is produced, it is difficult to separate the interests of producers and stakeholders. A lot of factors contribute to the welfare and health of animals in commercial production systems, i.e., housing system, feeding plans, and health programs [2]. Most of them, if not well managed, contribute to the onset of abnormal behaviors such as tail biting on pig farms, and this is an expensive and welfare-decreasing complication in the current modern swine industry. It is an economical and behavioral problem in the pork production system worldwide [3] and a systematic tail docking has been applied for decades to decrease the risk of biting. More recently, the European legislation in force, laying down minimum standards for the protection of pigs (EU Directive 120/2008), obliges farmers to not consider tail docking as a routine operation and to adopt the practice only where there is the evidence of injuries to the tails of other pigs. It is also mandatory that, before carrying out tail docking, measures shall be taken to prevent tail biting, taking into account environment, stocking densities, and management systems. In addition to this, the increased level of awareness of consumers and citizens in recent years has motivated several stakeholders of the production chain to develop a market-driven strategy on animal welfare standards [3], leading to the development of pork production from undocked pigs. Tail lesions derived from tail biting might be serious injuries, considered to be one of the most reliable indicators of animal welfare in pigs [4], and it is fundamental to monitor their occurrence during this transitional era towards long-tailed pig farming. Abattoir inspections provide a valuable tool for animal health and welfare monitoring, a source of data for epidemiological surveillance, and an opportunity for disease and lesions investigation that is more cost-effective for many pathologies and conditions than collecting data on farm [5,6]. Regarding tail lesions at slaughter, some studies have shown that such lesions, however mild, increased pleuritic and lung lesions, and that the risk increases with the severity of tail lesions [7]. However, other studies failed to find any correspondence [8,9,10]. The relationship between tail lesions and respiratory diseases is still uncertain [4,11] as both problems can have the same predisposing factors [4]. For example, poor housing conditions and management (i.e., incorrect airspeed in the barn, thermal discomfort, season, and low air quality due to the presence of dust and gas) have been shown to predispose the occurrence of both respiratory diseases and tail biting [4]. Tail biting and respiratory disease have also been described as the cause of similar effects on performances, such as lower carcass weight and higher carcass condemnation and/or carcass trimming [6,12]. In fact, other authors have reported a correlation between tail biting outbreaks and poor health conditions [7,13], leaving open the discussion about the relationship between respiratory disease and tail lesions. However, in the knowledge of the authors, literature is poor of investigations on the potential connection of these parameters between entire batches of pigs with undocked tails versus docked ones, as the majority of the studies mentioned above were performed on mainly docked animals. Moreover, gastric ulcers and quality of the thighs to produce Protected Designation of Origin (PDO) hams have not been investigated at all comparing docked and undocked pigs. Prevalence of gastric ulcers might increase after physical injuries, for example foot lesions or tail biting, as well as other secondary diseases that may result in the release of histamine due to an inflammation process [14]. Regarding PDO products, considering that only hams without defects are admitted to the market, it might be important from an economic point of view to investigate the eventual effect of undocked tails on defects of traumatic origin on the ham [15].

The study aimed at monitoring tail, pluck (lungs, pleurae, and liver), stomach, carcass, and thigh lesions or defects in slaughtering pigs belonging to either conventional docked batches or high-welfare standard batches from farms with the complete ban on tail docking.

## 2. Materials and Methods

### 2.1. Data Collection

Data were collected in one of the biggest Italian pig slaughterhouses (Società Cooperativa Agricola OPAS—Organizzazione Prodotto Allevatori Suini, Carpi, MO; Italy) for a period of 9 months (January–September 2021). Pigs designated for the PDO ham production (around 170 kg live body weight and 9 months of age) were transported to the slaughterhouse by trucks in batches of about 135 (from 130 to 140) animals reared in the same farm [16]. In each batch, an average of 100 animals (from 95 to 105) were selected for the evaluation. With biweekly visits, a total of 525 batches of pigs (52,500 animals) from intensive systems were evaluated: 442 batches from conventional farms adopting tail docking procedures (short, long-docking, or tipping) and 83 batches from high-welfare standard farms with the complete ban on tail docking. Batches from high-welfare standard farms were slaughtered early in the morning, followed by farms adopting tail docking procedures. Batches that were subjected to overnight lairage were noticed. The batches derived from 208 farms, with an average of 2.6 batches monitored for each farm. All the farms were located in Northern Italy, in the three regions more involved in the rearing of pigs for PDO ham (Lombardy, Emilia-Romagna, and Piedmont), where 77.2% of Italian commercial pig farms are located [17] and where last available data (2017) reported that 94.5% of farms rear pigs with docked tails [18]. Speed of the slaughter line was 480 animals/hour and the inspection was performed directly during the slaughter course. The time available for scoring each animal, determined by the speed of the line, was about 6–8 s.

Three veterinary expert evaluators were positioned in different platforms along the slaughter line to assign a score for each type of lesion (Table 1) and always conducted the inspection on the same position. Each veterinarian was previously trained through an introductory lecture regarding all the different scores in a 120-min oral presentation. The lecturer was a veterinarian expert in the slaughter lesions evaluation. After the lecture, the three auditors attended practical sessions on the slaughter line for 5 days in weekly scoring sessions, always under the guide of the expert.

### 2.2. Evaluation of the Pluck (Lungs, Pleurae, Liver, Heart)

The inspection of the pluck was carried out directly during the slaughter course from a dedicated platform placed right after the evisceration area. The examination of the lungs (mycoplasma-like lesions), pleurae, liver, and heart were performed by visual inspection and manual palpation of the organs, without any incisions. The score system is shown in Table 1 [5]. The scores were recorded using a voice recorder, and were transcribed in an Excel file for analysis.

### 2.3. Carcass (Skin) Lesions

Inspection of carcasses was performed directly during the slaughter course from a specific platform on the line after scalding and de-hairing of the carcass, and before dissection. Examination of external lesions was conducted by a visual inspection; the scores were registered in real time directly into an Excel file using a tablet.

To score traumatic lesions, the carcass was divided into three parts: the “posterior” region (hind legs and tail), the “trunk” (the ribs region), and the “anterior” region (the remaining area starting from the shoulders up to the front limbs, the head, and the ears). A 3-point scoring system was used for each of the three carcass regions to easily scan carcasses as they moved through the dressing line: score 0, up to one scratch or bite; score 1, from two to five scratches or bites; score 2, more than five scratches or bites, or any wound which penetrates the muscle [15]. In the case of lesions on ears and tail, the “chronic” (notches, necrosis, scab bites, and scars) or “fresh” (presence of blood or open wounds) classification was added.

### 2.4. Gastric Ulcers

Each stomach was opened by a machine about 15 min after jugulation of the animal by cutting along the large curvature and most of the gastric contents with a jet of water. Oesophago-gastric ulcers were classified using a 4-point scale proposed by Robertson [24] and Gottardo [25]: 0 = healthy, 1 = hyperkeratosis, 2 = erosion and/or mild ulcer, 3 = severe ulcer. The score system is shown in Table 1.

### 2.5. Defects of The Thigh

On the day after slaughter, all the separated thighs belonging to the batches previously scored for skin lesions were individually evaluated. The evaluation was performed by experts of the PDO ham according to PQI (Parma Quality Institute) standards [26]. Three traumatic (hematomas, muscular lacerations, tendon-bone lacerations) and three stress-related defects (pale soft exudative meat, petechial hemorrhaging, veining) were included for investigation as responsible for ham exclusion from the PDO market. In the case of more than one defect on the same ham, only the more severe and extensive one was considered.

### 2.6. Statistical Analysis

All the data were analyzed by using statistical software SAS (SAS Institute Inc., Cary, North Caroline, NC, USA, version 9.4). For each batch, the average score of each lesion was calculated, as well as the frequency of binary variables. Descriptive statistics of frequency of different scores for lesions were carried out (PROC UNIVARIATE). Normality of the data was checked by the Shapiro-Wilk test. For normally distributed data, the effect of tail docking on the different variables was assessed by ANOVA by using the Generalized linear model (PROC GLM). For non-normally distributed data, a log (ln + 1) transformation was applied. If normality was not achieved with transformation, the non-parametric Mann-Whitney test was used. The relationship between the prevalence or the average score of the different lesions was assessed at the batch level using Spearman’s rank correlation (PROC CORR).

## 3. Results

The frequency of undocked batches was 15.8%. The percentage of batches that arrived at the slaughterhouse the night before slaughtering and were subjected to overnight lairage was 53.3% for docked batches and 35.7% for undocked ones. The presence of tail lesions was only recorded in animals with undocked tails (44.0 ± 0.402 vs. 0.2 ± 0.2% compared to animals with docked tail, *p*-value < 0.001; minimum = 0.0%, maximum = 100.0%), with a prevalence of severe chronic lesions of 27.3% ± 0.032 (Table 2; minimum = 0.0%, maximum = 66.3%). On the contrary, docked animals showed more frequent ear lesions (9.6% ± 0.037 vs. 4.6% ± 0.019; *p* = 0.0001; minimum = 0.0%, maximum = 57.5%).

Severe lung lesions were found more frequently in undocked animals (9.2% ± 0.043 vs. 6.6% ± 0.011, *p* = 0.006; minimum = 0.0%, maximum = 48.4%), as well as gastric ulcers (26.1% ± 0.021 vs 20.3% ± 0.37, *p*-value = 0.006; minimum = 0.0%, maximum = 60.3%) and milk spot lesions in liver (17.8% ± 0.084 vs. 14.2% ± 0.302; *p* = 0.004; minimum = 1.0%, maximum = 99.0%). No difference was found for the pleura, skin lesions other than ears, and ham defects. Coefficient of variation among batches was <70.0% for all the statistically significant variables.

The descriptive analysis is reported as the average percentage of lesions detected for the different organs in relation to tail docking (Table 3 and Table 4). The presence of severe tail lesions (i.e., detected only in pigs with undocked tails) was correlated to mycoplasma-like lesions average score (chronic severe tail lesions: 0.135, *p* = 0.048; fresh tail lesions: 0.160, *p* = 0.019).

## 4. Discussion

The observation of lesions at the slaughterhouse is an investigative approach that has been previously adopted across Europe because it provides useful feedback to the farm. In fact, some information at the farm level is often unavailable for management purposes, for example the presence of visceral lesions in animals reaching the slaughter age without obvious clinical signs [27,28,29]. Several studies used the slaughterhouse as an important observatory for several kinds of lesions, including tail damage. However, the majority of these studies were performed on tail-docked animals [10,15,30] and investigations that compared these animals with pigs reared in farms that completely banned tail docking from their procedures are lacking. Analysis of the data collected in this study showed that animals with undocked tail had a higher frequency of tail lesions, both chronic and fresh, than docked ones. Actually, the percentage of docked animals with tail lesions was close to zero, opposite to the frequency of 44.0% of pigs with undocked tail showing damage. Results regarding the proportion of lesions in docked animals were in agreement with the low prevalence in heavy pigs reported by other authors [15,31,32], even if studies on lighter pigs in Europe showed higher frequencies and strong differences among countries, with values ranging from a prevalence of 72.5% [7] to 20–30% [33,34]. These differences were mainly explained by the large variability reported among different farms and batches [34], probably due to the unpredictable and multifactorial origin of tail biting [35]. Probably, considering that animals of older age are less likely to show tail lesions [31], the involvement of heavy pigs in the present study might be a factor reducing the observed frequency of tail lesions. Differently, the greater frequency of tail lesions observed in heavy pigs with undocked tail might confirm the findings of Scollo [31], who stated that the length of the tail after tail docking (short, long-docked, or tipped) might be responsible for different degrees of prevalence. In this case, the observation of both chronic and fresh lesions at slaughter put in question the age factor, which seems to be less relevant in the case of undocked tails. On the other hand, in batches of pigs with docked tails, a higher percentage of animals with ear lesions was observed, confirming results reported by Hunter et al. [36] and Bottacini [15] that suggested a substitution effect between tail and ear biting.

Two other important findings should be highlighted from the present study: the increased frequencies of pulmonary mycoplasma-like lesions and gastric ulcers in pigs with undocked tail. In the literature, several authors reported contrasting results on the possible association of tail lesions with other anatomopathological lesions, lung lesions in particular. Teixeira [30] reported a higher frequency of carcass condemnations due to pleurisy, pneumonia, and pleuropneumonia in the case of greater overall tail lesion score at batch level. Schrøder-Petersen and Simonsen [37] reported that the lungs are one of the organs most frequently affected by infection after the onset of a tail lesion. Similarly, Kritas and Morrison [13] reported an association between the severity of tail biting and the presence of mycoplasma-like lesions also at the individual level. However, other authors failed in identifying such association [10,38], probably because *Mycoplasma hyopneumoniae*, the bacteria mainly responsible for lung lesions recorded at slaughter, does not spread to the lungs via the blood [13]. Hence, the pathogenesis of mycoplasma-like lesions seems to be unrelated to tail trauma, which suggests that tail biting and mycoplasma-like lesions may share similar risk factors [4]. This hypothesis is supported by the correlation found in the present study between the presence of severe tail lesions and the mycoplasma-like lesions average score. However, it should be noted that mild mycoplasma-like lesions in lungs recorded at slaughter are often non-diagnostic of *M. hyopneumoniae* [39], and diagnosis based exclusively on macroscopic evaluation, especially in early or late stages of the disease, might lack reliability [40]. This might also suggest the involvement of different etiologies in the development of some mycoplasma-like lesions. In fact, the result of the adhesion of *M. hyopneumoniae* to the respiratory epithelium and its stimulation is a prolonged inflammatory reaction, with the suppression and modulation of the innate and adaptive immune responses of the host. Then, infected animals become more susceptible to secondary infections of other respiratory pathogens [41]. In situ hybridization results reported by Amass [40] suggested that the destruction of pulmonary macrophages after infection by *M. hyopneumoniae* is an indication of its direct pathogenic effect. Such changes adversely affect the respiratory defense mechanisms of the host and commonly lead to secondary bacterial infections with *Pasteurella multocida* and *Actinobacillus pleuropneumoniae*, but also with *Actinomyces pyogenes*, streptococci, and staphylococci [42,43]. Some of these latter pathogens, in particular Streptococcus spp. and Staphylococcus spp., may originate from a systemic infection most often secondary to bacterial infections of skin traumas, first of all tail biting lesions [44], that cause septicemia and generalized hematogenous spread of bacteria [45]. Due to this mechanism, previous studies regarding tail-docked pigs [33] have reported a close association between tail lesions and the presence of abscesses on the carcass and/or in the lungs, or even with the development of pyemia. In the present study, no difference was found in lung abscesses between docked and undocked pigs, but the tendency to a higher prevalence of lung tissue consolidation might support this hypothesis. Regarding the frequency of mycoplasma-like lesions observed in the present study, results are lower than those reported by the same authors in 2017 [5] and by older research in heavy pigs [46], confirming the lower frequency reported in a more recent study [10]. These authors suggested that the differences between the studies over time might be imputable to the wide diffusion of vaccination against *M. hyopneumoniae* in Italy in recent years and to its protective effects. Differently, the average SPES score is similar to those reported in previous studies [5,10,16,47].

In the present study, undocked batches showed a higher percentage of gastric ulcers compared to docked ones. It might confirm the hypothesis that physical injuries, for example tail biting but also foot lesions or other secondary diseases, may cause a release of histamine after an inflammation process, resulting in increasing prevalence of gastric ulcers [14]. Also, lung lesions might be a factor increasing gastric ulcers [25], not only for the increased levels of histamine due also to infection with respiratory pathogens, but possibly also as a consequence of inappetence due to illness [14]. Another hypothesis might be that the pathogenesis of gastric ulcers, as well as the onset of mycoplasma-like lesions, might be unrelated to tail trauma but they may share similar risk factors. In fact, several factors that are generally considered important for pig welfare on farm also showed an association with gastric ulcers. For example, the presence of solid flooring, straw, and other environmental enrichments, as well as the absence of mixing animals, are all protective factors that decrease the risk, suggesting the potential for psychological and physical stressors to affect gastric ulcers [25]. An important predisposing factor to gastric ulcers is the overnight lairage at slaughter [25], but both docked and undocked groups in the present study had a percentage of batches that were subjected to lairage and to the consequent fasting time, reducing a possible confounding effect. Regarding the frequency of gastric ulcers observed in the present study, results are in agreement with Gottardo et al. [25].

In the present study, batches from farms abandoning tail docking showed lower frequency of liver lesions. However, the authors consider this finding as minor due to possible confounding factors in the data collection. In fact, most of the undocked batches came from farms voluntarily adopting high-welfare standard protocols of production required by specific supplying chains. Among the requirements is a greater attention to antiparasitic treatments on animals. Unfortunately, data regarding drug use were not collected and the hypothesis could not be confirmed. However, it would be important to note that Gottardo [25] found that when antiparasitic treatments were not provided, risk of gastric ulcers increased by a factor of three. Tarakdjian [48] described the high welfare standard farms rearing heavy pigs as a type of production characterized by a higher level of veterinary support to farmers and more intensive and frequent education programs for farm personnel on good management practices (including antiparasitic treatments and drug usage), biosecurity, and animal welfare [48]. Frequency of liver lesions observed in the present study are lower than those reported by Scollo [5], supporting the hypothesis of a greater attention to antiparasitic treatments in several of the farms involved.

A minor importance was attributed also to the higher frequency of lesions in the anterior part of the carcass of undocked batches, showing only a statistical tendency and few biological causalities with long tails. Scratches on the anterior part of the carcass are usually attributed to fighting behavior before slaughter, which are more likely to take place when unfamiliar pigs are mixed together during or after loading/unloading procedures [15]. Unfortunately, data regarding groups management on farm or at slaughter were not available.

The last result was the absence of significant increase of ham defects in undocked batches. This is very important for the ham market because PDO production schemes, which are designed to produce typical charcuterie products, admit in their market only high-quality hams without defects [26]; this leads to a higher economic impact of thigh lesions compared to other production systems, considering that PDO thighs have a 20–40% of extra value [15]. The absence of differences in the frequency of ham defects in pigs with undocked tail reassures this specific market about the ban of tail docking. The doubt was legitimate as other kinds of economic losses have been reported by other authors in tail-bitten animals, in particular frequency of carcass condemnations and a lower carcass weight associated to the severity of tail lesions, where considerable financial losses were identified and primarily associated to lesions caused by tail biting [7]. For the specific Italian area, data obtained in the present study might be of interest as the Ministry of Health recently recommended, for the triennium 2020–2023, that farmers rear pilot groups of pigs with intact tails (>1.5% in farms with >2000 pigs; >3.0% in smaller farms), defining progressive goals which have the potential to drive the production towards the abandonment of routine tail docking and the accomplishment of the European standards [49].

## 5. Conclusions

This monitoring study on several visceral lesions and product defects detected on docked or batches of heavy pigs with undocked tail confirms the importance of the slaughterhouse as an investigation center not only for notifiable disease or health disorders, but also for animal welfare. Data from the slaughter line can be collected and used to adopt preventive measures that could positively affect sanitary and welfare management on-farm. The higher prevalence of tail lesions on batches with undocked tail compared to docked ones shown in the present study gave rise to the suggestion that more and alternative efforts to manage animals with undocked tail are needed towards a more acceptable and welfare-friendly farming system. In particular, farms rearing pigs with undocked tail that successfully showed no (or few) tail lesions should be further investigated as an example of virtuous management for the other farms. Long tail lesions might be suggested as an iceberg indicator for other pathological conditions, such as the presence of mycoplasma-like lesions in lungs and gastric ulcers, even if it is still not clear if tail lesions share the same predisposing factor with lung lesions and gastric ulcers or if tail lesions might have a role in the causality and onset of the other conditions. Eventual causal association among lesions in these organs should be explored. The absence of increased frequency in ham defects in batches with undocked tail might reassure PDO producers on the quality of their product in the case of a tail docking ban on-farm.

## Figures and Tables

**Table 1 animals-13-00305-t001:** Scoring system used for the assessment of pleural, liver, and lung lesions at slaughter from January to September. A total number of 525 batches of pigs (135 pigs per batch, around 170 kg weight) were monitored. Reprinted and slightly adapted with permission from Elsevier [5]. Copyright 2017, Elsevier B.V.

Injuries	Scale	Description
**Lungs**		
Lung score (mycoplasma-like lesions)	0–24	Pulmonary lesions (mycoplasma-like lesions), often due to *Mycoplasma Hyopneumoniae*: purple to grey rubbery consolidation, increased firmness, failure to collapse and edema were scored according to Madec’s grid [19]. Cranial, medial, and diaphragmatic lobes of both lungs were scored each from 0 to 4.
Absence of lesions Severe lesions Scars	0–1 0–1 0–1	Lungs receiving score 0 in all the evaluated lobes. Lungs with a score ≥5/24. Presence of retracted tissue from recovered enzootic pneumonia-like lesions, with thickened interlobular purple to grey connective tissue.
Abscesses Consolidations	0–1 0–1	Presence of at least one abscess in the evaluated lobes. Pulmonary lesions complicated by secondary bacterial pathogens (e.g., Pasteurella spp., Bordetella spp.). Lesions appeared firmer and heavier than mycoplasma-like lesions. On a cut surface, lesions were mottled by arborized clusters of grey to white distended alveoli with exudation. A mucopurulent exudate could be expressed from the upper respiratory tract [20].
Lobular/chessboard pattern lesions	0–1	Presence of scattered multifocal spots of purple to grey discoloration indicative of probable co-existence of respiratory viruses (e.g., porcine reproductive and respiratory virus, porcine circovirus, influenza virus) and/or Mycoplasma spp. or foreign body (e.g., dust/particulate matter) [21].
**Pleura**		
Pleura score (SPES score)	0–4	SPES grid [22]. 0: absence of pleural lesions; 1: cranioventral pleuritis and/or pleural adherence between lobes or at ventral border of lobes; 2: dorsocaudal unilateral focal pleuritis; 3: bilateral pleuritis of type 2 or extended unilateral pleuritis (at least 1/3 of one diaphragmatic lobe); 4: severely extended bilateral pleuritis (at least 1/3 of both diaphragmatic lobes).
Severe lesions Sequestra	0–1 0–1	Pleura with a SPES score ≥3. Presence of at least one sequestra in the lungs: abundant fibrin on the surface, and hemorrhagic, necrotic parenchyma; in case of chronicity, the remaining cavitated necrotic foci after partial resolution are surrounded by scar tissue. Often associated with *Actinobacillus pleuropneumoniae* infection [23].
**Liver**		
Liver score	1–3	Scoring based on the presence of lesions due to *Ascaris suum* (milk spots) and its migration. 1: no lesions or less than 4 lesions; 2: from 4 to 10 lesions; 3: more than 10 lesions.
Severe lesions Total lesions	0–1 0–1	Livers with a score of 3. Livers with a score ≥2.
**Stomach**		
Gastric ulcers	0–3	0: healthy; 1: hyperkeratosis; 2: erosion and/or mild ulcer; 3: severe ulcer.
Absence of lesions	0–1	Stomachs that received score 0.
Total ulcers (mild to severe, %)	0–1	Stomachs that received score 2–3.

**Table 2 animals-13-00305-t002:** Frequency of tail and ear lesions (±standard error) in relation to tail docking: data expressed as affected subjects/batch.

Parameters	Docked Tail	Undocked Tail	*p*-Value
Absence of tail lesions (%)	99.8 ± 0.2	56.0 ± 0.402	<0.001
Mild and fresh tail lesions (%)	0.1 ± 0.1	8.8 ± 0.005	<0.001
Mild and chronic tail lesions (%)	0.1 ± 0.1	0.9 ± 0.002	<0.001
Severe and fresh tail lesions (%)	0.0 ± 0.0	7.2 ± 0.008	<0.001
Severe and chronic tail lesions (%)	0.0 ± 0.0	27.3 ± 0.032	<0.001
Ear lesions (%)	9.6 ± 0.037	4.6 ± 0.019	0.001

**Table 3 animals-13-00305-t003:** Average percentage of lesions (±standard error) observed in different organs of the pluck (lungs, pleurae, liver, heart) and gastric ulcers. Only *p*-values lower than 0.090 are reported.

Parameters	Docked tail	Undocked Tail	*p*-Value
**Pluck lesions**			
Lung’s mycoplasma-like lesions			
Absence of lesions (%)	59.4 ± 0.811	54.4 ± 0.815	0.017
Severe lesions (%)	6.6 ± 0.011	9.2 ± 0.043	0.006
Average score	1.12 ± 0.14	1.44 ± 0.023	0.001
Scars (%)	10.67 ± 0.49	11.87 ± 0.86	ns
Abscesses (%)	0.89 ± 0.10	0.73 ± 0.19	ns
Consolidations (%)	0.16 ± 0.04	0.28 ± 0.07	0.082
Lobular/chessboard pattern lesions (%)	2.25 ± 0.19	2.20 ± 0.34	ns
Pleural injuries			
Severe lesions (%)	12.55 ± 0.60	12.83 ± 1.07	ns
Average SPES score	0.89 ± 0.03	0.97 ± 0.05	ns
Liver injuries			
Severe lesions (%)	5.4 ± 0.015	4.1 ± 0.014	0.021
Total lesions (mild to severe, %)	17.8 ± 0.084	14.2 ± 0.302	0.004
Average score	1.23 ± 0.053	1.18 ± 0.102	0.005
Pericarditis (%)	5.43 ± 0.33	6.36 ± 0.59	ns
**Gastric ulcers**			
Absence of ulcers (%)	10.06 ± 1.58	7.16 ± 2.47	ns
Total ulcers (mild to severe, %)	20.3 ± 1.46	26.1 ± 2.29	0.006
Average score	1.13 ± 0.03	1.22 ± 0.05	0.030

ns: not statistically significant (*p*-value > 0.090).

**Table 4 animals-13-00305-t004:** Average percentage of lesions (±standard error*) observed in different parts of the carcass (anterior, trunk, posterior) and ham defects in relation to tail docking. Only *p*-values lower than 0.090 are reported.

Parameters	Docked Tail	Undocked Tail	*p*-Value
**Carcass lesions**			
Anterior region			
Absence of injuries (%)	80.1 ± 0.890	74.8 ± 0.972	0.071
Mild injury (%)	15.4 ± 0.789	18.9 ± 0.654	0.065
Severe injury (%)	4.5 ± 0.476	6.4 ± 0.782	0.059
Trunk			
Absence of injuries (%)	85.3 ± 0.803	82.0 ± 0.764	ns
Mild injury (%)	12.2 ± 0.912	12.3 ± 0.546	ns
Severe injury (%)	3.2 ± 0. 510	4.8 ± 0.128	ns
Posterior region			
Absence of injuries (%)	77.5 ± 0.06	75.5 ± 0.023	0.073
Mild injury (%)	18.1 ± 0.50	20.2 ± 0.902	ns
Severe injury (%)	3.2 ± 0.07	3.6 ± 0.891	ns
**Ham defects**			
Traumatic defects			
Hematomas (%) ^1^	3.3 ± 2.1	3.0 ± 1.9	ns
Muscular lacerations (%) ^1^	0.9 ± 1.1	1.1 ± 1.2	ns
Tendon-bone lacerations (%) ^1^	0.3 ± 0.5	0.4 ± 1.0	ns
Stress related defects			
PSE (%) ^1^	0.2 ± 0.4	0.3 ± 0.8	ns
Petechial hemorrhaging (%) ^1^	0.6 ± 1.0	0.7 ± 1.0	ns
Veining (%) ^1^	5.0 ± 4.6	4.7 ± 4.0	ns

^1^ Data analyzed with a non-parametric Mann Whitney test; standard error is replaced by standard deviation. PSE: Pale Soft and Exudative meat. Ns: not statistically significant (*p*-value > 0.090).

## Data Availability

All the obtained data are available within the article.

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
