# Peer review of "Undocked Tails, Mycoplasma-like Lesions and Gastric Ulcers in Slaughtering Pigs: What Connection?"

_animals, 2023, doi:10.3390/ani13020305_

Round 1

Reviewer 1 Report

This is an interesting study on an important societal topic, the welfare of raising undocked pigs. The metholodogy is clear and the results also. The discussion follows logically from the results and nicely compares with other publications done. 

I only have small remarks on the study and some general questions: 

- the pigs were distributed in batches of undocked pigs and batches of docked pigs. However, docking can happen in degrees, docking a large part of the tail or a smaller part or tipping the tail. Were all pigs that were docked in any or the other way in the 'docked group'? 

- the correlation was looked at between different types of lesions in docked and undocked pigs. However, could you also see a relation in e.g. undocked pigs with tail lesions with the other legions observed? Or docked pigs with ear lesions and other lesions? 

- Would be nice to write something more about the variability between the batches. 

- Sometimes the term undocked pigs and sometimes the term long tails is used both in the text as in the tables (e.g. table 2 and 3). I suggest to use the same wording for consistency. 

I added some extra suggestions in the attached file. 

Author Response

This is an interesting study on an important societal topic, the welfare of raising undocked pigs. The metholodogy is clear and the results also. The discussion follows logically from the results and nicely compares with other publications done. 

Au: we thank the reviewer for the precious comments. In particular, thanks for highlighting some our mistakes. All the suggestions have been approved (see also attached PDF). Below answers to the general question.

I only have small remarks on the study and some general questions: 

- the pigs were distributed in batches of undocked pigs and batches of docked pigs. However, docking can happen in degrees, docking a large part of the tail or a smaller part or tipping the tail. Were all pigs that were docked in any or the other way in the 'docked group'? 

Au: yes, all the docked pigs (short-, long-docked, or only tipped) were in the docked group. Only intact tails were considered “long tails”. In order to reduce the risk of misunderstanding, in the entire text we have replaced most of the term “long tails” with “undocked tails”. A short sentence has been added in the m&m (L103).

- the correlation was looked at between different types of lesions in docked and undocked pigs. However, could you also see a relation in e.g. undocked pigs with tail lesions with the other legions observed? Or docked pigs with ear lesions and other lesions? 

Au: as tail lesions were detected quite only in undocked pigs, correlation in e.g. undocked pigs with tail lesions with the other legions observed is quite the same as reported. A short sentence has been added to clarify it (L180). Differently, relation in e.g. undocked pigs with tail lesions with the other legions observed were not tested. However, we have re-run the program in order to better investigate it, but no adjunctive information arose (no correlation between ear lesions and other lesions in docked pigs, as well as when we considered both docked and undocked pigs). So, no other information has been added in the text.

- Would be nice to write something more about the variability between the batches. 

Au: A comment on coefficient of variation has been added in the results (L180-1). To improve the data description, we added information regarding minimum and maximum values in the statistically significant variables in the text (L171-9). This has leaded to a new short sentence in the conclusion (L332-5) in the conclusions.

- Sometimes the term undocked pigs and sometimes the term long tails is used both in the text as in the tables (e.g. table 2 and 3). I suggest to use the same wording for consistency. 

Au: in the entire text we have replaced most of the term “long tails” with “undocked tails”.

I added some extra suggestions in the attached file. 

Au: all the suggestions have been applied.

Reviewer 2 Report

The manuscript entitled ‘Undocked Tails, Mycoplasma-Like Lesions and Gastric Ulcers in Slaughtering Pigs: What Connection?’ by Scollo et al tackles the issue of using abattoir-based measures to determine the health and welfare status in either docked or undocked heavy pigs.  

The authors have reported that in pigs with intact tails the presence of tail, lung and gastric lesions were more frequently found compared to pigs with not-intact tails, suggesting that tail lesions could be used as indicators of imbalanced welfare and health on-farms conditions. 

Overall, the manuscript is well-written, the aims are clear, and the conclusion supported by the results. However, I reckon that the paper could be improved especially in the material and methods sections as well in the discussion section. 

Here the authors might find some points that, in my opinion, require their attention: 

L53: It would be interesting to describe the current situation in Italy in order to highlight the difference between the prevalence of farms holding pigs which are regularly tail docked and farms holding pigs not routinely tail docked. 

L95: Includes the year when the data collection has been completed. 

L110: Please provide a more detailed description of the training sessions (e.g., were they always conducted by the same trainer? were they performed exclusively during the slaughter course or also though the use of images or videos? 

L139: I reckon that 10 minutes after the jugulation would be a really short time for assessing the stomachs, considering the standard operating procedures in modern pig slaughterhouses: please check this sentence. 

L150-151: It is not clear to me the system used to ascertain which one of the stress-related defects would be considered as the worst in the event of contemporary presence of more than once of these (e.g., is veining worst the PSE? Why?) 

L217: are there other references supporting the fact that pigs with docked tails tend to have more damage on the ear due to this substitution effect? 

L232: Do the authors have any information concerning the management procedures conducted on the farms from which the pigs assessed in this study originate? This could also be beneficial with regard to the occurrence of gastric lesions, as the type of feeding is recognized as probably the most important risk factor for these conditions.  

L263: Did the authors explored the difference in gastric lesions occurrence with regard to the feed withdrawal period time before slaughtering? Did the pigs assessed in the study spend any time in the lairage?

Author Response

The manuscript entitled ‘Undocked Tails, Mycoplasma-Like Lesions and Gastric Ulcers in Slaughtering Pigs: What Connection?’ by Scollo et al tackles the issue of using abattoir-based measures to determine the health and welfare status in either docked or undocked heavy pigs.  

The authors have reported that in pigs with intact tails the presence of tail, lung and gastric lesions were more frequently found compared to pigs with not-intact tails, suggesting that tail lesions could be used as indicators of imbalanced welfare and health on-farms conditions. 

Overall, the manuscript is well-written, the aims are clear, and the conclusion supported by the results. However, I reckon that the paper could be improved especially in the material and methods sections as well in the discussion section. 

Au: we thank the reviewer for the precious comments. Below answers to the question.

Here the authors might find some points that, in my opinion, require their attention: 

L53: It would be interesting to describe the current situation in Italy in order to highlight the difference between the prevalence of farms holding pigs which are regularly tail docked and farms holding pigs not routinely tail docked. 

Au: data regarding the Italian prevalence of docked and undocked farms has been added in the paragraph describing the study area (L111-2). Unfortunately, the last reference is not really updated (2018 on data collected in 2017), but more recent data are not available. However, to better describe the current Italian situation in more recent time, a sentence has been added in the discussion (L324-32).

L95: Includes the year when the data collection has been completed. 

Au: done

L110: Please provide a more detailed description of the training sessions (e.g., were they always conducted by the same trainer? were they performed exclusively during the slaughter course or also though the use of images or videos? 

Au: a short paragraph has been added in the text (L116-24).

L139: I reckon that 10 minutes after the jugulation would be a really short time for assessing the stomachs, considering the standard operating procedures in modern pig slaughterhouses: please check this sentence. 

Au: To be more prudent, time has been increased up to 15 min. However, we are sure of the accuracy of the information. A similar timing was reported by Gottardo et al. (2017) for the same Italian slaughterhouse.

L150-151: It is not clear to me the system used to ascertain which one of the stress-related defects would be considered as the worst in the event of contemporary presence of more than once of these (e.g., is veining worst the PSE? Why?) 

Au: the classification is based on the extension (and severity) of the defect. A clarification has been added in the text.

L217: are there other references supporting the fact that pigs with docked tails tend to have more damage on the ear due to this substitution effect? 

Au: Another reference has been added.

L232: Do the authors have any information concerning the management procedures conducted on the farms from which the pigs assessed in this study originate? This could also be beneficial with regard to the occurrence of gastric lesions, as the type of feeding is recognized as probably the most important risk factor for these conditions.  

Au: unfortunately, as this is not a retrospective study, we do not have information about the management of the 208 farms involved. However, to address the causes of gastric ulcers was not the aim of the study, and other studies are needed to better investigate the possible relationship with undocked tails.

L263: Did the authors explored the difference in gastric lesions occurrence with regard to the feed withdrawal period time before slaughtering? Did the pigs assessed in the study spend any time in the lairage?

Au: information regarding the lairage has been added, including the percentage of batches that were subjected to it (L108; L178-80). A sentence has been added also in the discussion (L297-300). Thanks for this important advice.